# Learning to Represent Programs with Property Signatures

**Augustus Odena, Charles Sutton**
Google Research
{augustusodena,charlessutton}@google.com

## Abstract

We introduce the notion of property signatures, a representation for programs and program specifications meant for consumption by machine learning algorithms. Given a function with input type $\tau_{in}$ and output type $\tau_{out}$, a property is a function of type: $(\tau_{in}, \tau_{out}) \rightarrow$ `Bool` that (informally) describes some simple property of the function under consideration. For instance, if $\tau_{in}$ and $\tau_{out}$ are both lists of the same type, one property might ask 'is the input list the same length as the output list?'. If we have a list of such properties, we can evaluate them all for our function to get a list of outputs that we will call the property signature. Crucially, we can 'guess' the property signature for a function given only a set of input/output pairs meant to specify that function. We discuss several potential applications of property signatures and show experimentally that they can be used to improve over a baseline synthesizer so that it emits twice as many programs in less than one-tenth of the time.

## 1 Introduction

Program synthesis is a longstanding goal of computer science research (Manna & Waldinger, 1971; Waldinger et al., 1969; Summers, 1977; Shaw; Pnueli & Rosner, 1989; Manna & Waldinger, 1975), arguably dating to the 1940s and 50s (Copeland, 2012; Backus et al., 1957). Deep learning methods have shown promise at automatically generating programs from a small set of input-output examples (Balog et al., 2016; Devlin et al., 2017; Ellis et al., 2018b; 2019b). In order to deliver on this promise, we believe it is important to represent programs and specifications in a way that supports learning. Just as computer vision methods benefit from the inductive bias inherent to convolutional neural networks (LeCun et al., 1989), and likewise with LSTMs for natural language and other sequence data (Hochreiter & Schmidhuber, 1997), it stands to reason that ML techniques for computer programs will benefit from architectures with a suitable inductive bias.

We introduce a new representation for programs and their specifications, based on the principle that *to represent a program, we can use a set of simpler programs*. This leads us to introduce the concept of a property, which is a program that computes a boolean function of the input and output of another program. For example, consider the problem of synthesizing a program from a small set of input-output examples. Perhaps the synthesizer is given a few pairs of lists of integers, and the user hopes that the synthesizer will produce a sorting function. Then useful properties might include functions that check if the input and output lists have the same length, if the input list is a subset of the output, if element 0 of the output list is less than element 42, and so on.

The outputs of a set of properties can be concatenated into a vector, yielding a representation that we call a *property signature*. Property signatures can then be used for consumption by machine learning algorithms, essentially serving as the first layer of a neural network. In this paper, we demonstrate the utility of property signatures for program synthesis, using them to perform a type of premise selection as in Balog et al. (2016). More broadly, however, we envision that property signatures could be useful across a broad range of problems, including algorithm induction (Devlin et al., 2017), improving code readability (Allamanis et al., 2014), and program analysis (Heo et al., 2019).

More specifically, our contributions are:

- We introduce the notion of property signatures, which are a general purpose way of featurizing both programs and program specifications (Section 3).

- We demonstrate how to use property signatures within a machine-learning based synthesizer for a general-purpose programming language. This allows us to automatically learn a useful set of property signatures, rather than choosing them manually (Sections 3.2 and 4).

- We show that a machine learning model can predict the signatures of individual functions given the signature of their composition, and describe several ways this could be used to improve existing synthesizers (Section 5).

- We perform experiments on a new test set of 185 functional programs of varying difficulty, designed to be the sort of algorithmic problems that one would ask on an undergraduate computer science examination. We find that the use of property signatures leads to a dramatic improvement in the performance of the synthesizer, allowing it to synthesize over *twice as many programs in less than one-tenth of the time* (Section 4). An example of a complex program that was synthesized only by the property signatures method is shown in Listing 1.

For our experiments, we created a specialized programming language, called Searcho[1] (Section 2), based on strongly-typed functional languages such as Standard ML and Haskell. Searcho is designed so that many similar programs can be executed rapidly, as is needed during a large-scale distributed search during synthesis. We release[2] the programming language, runtime environment, distributed search infrastructure, machine learning models, and training data from our experiments so that they can be used for future research.

```
1  fun unique_justseen(xs :List<Int>) -> List<Int> {
2    let triple = list_foldl_<Int, (List<Int>, Int, Bool)>(
3      xs,
4      (nil<Int>, 0, _1),
5      \(list_elt, (acc, last_elt, first)){
6        cond_(or_(first, not_equal_(list_elt, last_elt)),
7          \{(cons_(list_elt, acc), list_elt, _0)},
8          \{(acc                , list_elt, _0)})
9      });
10   list_reverse_(#0(triple))
11 };
```

Listing 1: A program synthesized by our system, reformatted and with variables renamed for readability. This program returns the sub-list of all of the elements in a list that are distinct from their previous value in the list.

## 2    PROGRAMMING BY EXAMPLE AND THE SEARCHO LANGUAGE

In Inductive Program Synthesis, we are given a specification of a program and our goal is to synthesize a program meeting that specification. Inductive Synthesis is generally divided into Programming by Example (PBE) and Programming by Demonstration (PBD). This work is focused on PBE. In PBE, we are given a set of input/output pairs such that for each pair, the target program takes the input to the corresponding output. Existing PBE systems include Winston (1970), Menon et al. (2013), and Gulwani (2011). A PBE specification might look like Listing 2:

```
1  io_pairs = [(1, 1), (2, 4), (6, 36), (10, 100)]
```

Listing 2: An example PBE specification.

for which a satisfying solution would be the function squaring its input. Arbitrarily many functions satisfy this specification. It is interesting but out of scope[3] to think about ways to ensure that the synthesis procedure recovers the 'best' or 'simplest' program satisfying the specification.

Much (though not all) work on program synthesis is focused on domain specific languages that are less than maximally expressive (Gulwani, 2011; Balog et al., 2016; Wang et al., 2017; Alur et al.,

---

[1]Searcho is heavily based on code written by Niklas Een, which is available at https://github.com/tensorflow/deepmath/tree/master/deepmath/zz/CodeBreeder

[2]Available at https://github.com/brain-research/searcho

[3]Though note that in this work and in prior work, the search procedure used will tend to emit 'shorter' programs first, and so there is an Occam's-Razor-type argument (Spade & Panaccio, 2019) to be made that you *should* get this for free.

2015). We would like to focus on the synthesis of programs in a Turing complete language, but this presents technical challenges: First, general purpose languages such as C++ or Python are typically quite complicated and sometimes not fully specified; this makes it a challenge to search over partial programs in those languages. Second, sandboxing and executing code written in these languages is nontrivial. Finally, searching over and executing many programs in these languages can be quite slow, since this is not what they were designed for.

For these reasons, we have created a general-pupose, Turing complete programming language and runtime. The programming language is called Searcho and it and its runtime have been designed specifically with program synthesis in mind. The language can roughly be thought of as a more complicated version of the simply typed lambda calculus or as a less complicated version of Standard ML or OCaml.[4] Searcho code is compiled to bytecode and run on the Searcho Virtual Machine. Code is incrementally compiled, which means that the standard library and specification can be compiled once and then many programs can be pushed on and popped off from the stack in order to check them against the specification. Searcho is strongly typed with algebraic datatypes (Pierce & Benjamin, 2002)[5] Searcho includes a library of 86 functions, all of which are supported by our synthesizer. This is a significantly larger language and library than have been used in previous work on neural program synthesis.

We have also implemented a baseline enumerative synthesizer. The main experiments in this paper will involve plugging the outputs of a machine learning model into the configuration for our baseline synthesizer to improve its performance on a set of human-constructed PBE tasks.

## 3 PROPERTY SIGNATURES

Consider the PBE specification in Listing 3:

```
1 io_pairs = [
2   ([1, 2345, 34567],    [1, 2345, 34567, 34567, 2345, 1]),
3   ([True, False],       [True, False, False, True]),
4   (["Batman"],          ["Batman", "Batman"]),
5   ([[1,2,3], [4,5,6]], [[1,2,3], [4,5,6], [4,5,6], [1,2,3]])
6 ]
```

Listing 3: An example PBE Specification.

We can see that the function concatenating the input list to its reverse will satisfy the specification, but how can we teach this to a computer? Following Balog et al. (2016) we take the approach of training a machine learning model to do premise selection for a symbolic search procedure. But how do we get a representation of the specification to feed to the model? In Balog et al. (2016), the model acts only on integers and lists of integers, constrains all integers to lie in $[-256, 256]$, has special-case handling of lists, and does not deal with polymorphic functions. It would be hard to apply this technique to the above specification, since the first example contains unbounded integers, the second example contains a different type than the first[6], and the third and fourth examples contain recursive data structures (lists of characters and lists of integers respectively).

Thankfully, we can instead learn a representation that is composed of the outputs of multiple other programs running on each input/output pair. We will call these other programs properties. Consider the three properties in Listing 4.

```
1 all_inputs_in_outputs ins outs    = all (map (\x -> x in outs) ins)
2 ouputs_has_dups ins outs          = has_duplicates (outs)
3 input_same_len_as_output ins outs = (len ins) == (len outs)
```

Listing 4: Three function projections that can act on the specification from Listing 3.

Each of these three programs can be run on all 4 of the input output pairs to yield a `Boolean`. The first always returns True for our spec, as does the second. The third always returns False on the given examples, although note that it would return True if the examples had contained the implicit base case of the empty list. Thus, we can write that our spec has the 'property signature' [True, True, False].

---

[4]In this paper, we will present illustrative programs in Haskell syntax to make them more broadly readable. Searcho programs will be presented in Searcho syntax, which is similar.

[5]Types have been shown to substantially speed up synthesis. See e.g. Figure 6 of Feser et al. (2015).

[6]So any function satisfying the spec will be parametrically polymorphic.

How is this useful? From the first property we can infer that we should not throw away any elements of the input list. From the third we might guess that we have to add or remove elements from the input list. Finally, the second might imply that we need to create copies of the input elements somehow. This does not narrow our search down all the way, but it narrows it down quite a lot. Since the properties are expressed in the same language as the programs we are synthesizing, we can emit them using the same synthesizer. Later on, we will describe how we enumerate many random properties and prune them to keep only the useful ones. The property signatures that we consider in our experiments contain thousands of values.

Since the output of these properties is either always True, always False, or sometimes True and sometimes False, a neural network can learn embeddings for those three values and it can be fed a vector of such values, one for each applicable property, as the representation of a program specification.

### 3.1 Abstracting Properties into Signatures

Now we describe our representation for a program $f :: \tau_{in} \to \tau_{out}$. Each property is a program $p :: (\tau_{in}, \tau_{out}) \to \texttt{Bool}$ that represents a single "feature" of the program's inputs and outputs which might be useful for its representation.[7] In this section, we assume that we have determined a sequence $P = [p_1 \ldots p_n]$ of properties that are useful for describing $f$, and we wish to combine them into a single representation of $f$. Later, we will describe a learning principle for choosing relevant properties.

We want the property signature to summarize the output of all the properties in $P$ over all valid inputs to $f$. To do this, we first extend the notion of property to a set of inputs in the natural way. If $S$ is a set of values of type $\tau_{in}$ and $p \in P$, we define $p(S) = \{p(x, f(x)) \mid x \in S\}$. Because $p(S)$ is a set of booleans, it can have only three possible values, either $p(S) = \{\mathsf{True}\}$, or $p(S) = \{\mathsf{False}\}$, or $p(S) = \{\mathsf{True}, \mathsf{False}\}$, corresponding respectively to the cases where $p$ is always true, always false, or neither. To simplify notation slightly, we define the function $\Pi$ as $\Pi(\{\mathsf{True}\}) = \mathsf{AllTrue}$, $\Pi(\{\mathsf{False}\}) = \mathsf{AllFalse}$, and $\Pi(\{\mathsf{True}, \mathsf{False}\}) = \mathsf{Mixed}$. Finally, we can define the *property signature* $\mathrm{sig}(P, f)$ for a program $f$ and a property sequence $P$ as

$$\mathrm{sig}(P, f)[i] = \Pi(p_i(V(\tau_{in}))),$$

where $V(\tau_{in})$ is the possibly infinite set of all values of type $\tau_{in}$.

Computing the property signature for $f$ could be intractable or undecidable, as it might require proving difficult facts about the program. Instead, in practice, we will compute an *estimated property signature* for a small set of input-output pairs $S_{io}$. The estimated property signature summarizes the actions of $P$ on $S_{io}$ rather than on the full set of inputs $V(\tau_{in})$. Formally, the estimated property signature is

$$\widehat{\mathrm{sig}}(P, S_{io})[i] := \Pi(\{p_i(x_{in}, x_{out}) \mid (x_{in}, x_{out}) \in S_{io}\}). \tag{1}$$

This estimate gives us an under-approximation of the true signature of $f$ in the following sense: If we have $\widehat{\mathrm{sig}}(P, S) = \mathsf{Mixed}$, we must also have $\mathrm{sig}(P, f) = \mathsf{Mixed}$. If $\widehat{\mathrm{sig}}(P, S) = \mathsf{AllTrue}$, then either $\mathrm{sig}(P, f) = \mathsf{AllTrue}$ or $\mathrm{sig}(P, f) = \mathsf{Mixed}$, and similarly with $\mathsf{AllFalse}$. Estimated property signatures are particularly useful for synthesis using PBE, because we can compute them from the input-output pairs that specify the synthesis task, without having the definition of $f$. Thus we can use estimated property signatures to 'featurize' PBE specifications for use in synthesis.

### 3.2 Learning Useful Properties

How do we choose a set of properties that will be useful for synthesis? Given a training set of random programs with random input/output examples, we generate many random properties. We then prune the random properties based on whether they distinguish between any of the programs. Then, given a test suite of programs, we do an additional pruning step: among all properties that give the same value for every element of the test suite, we keep the shortest property, because of Occam's razor considerations. Given these 'useful' properties, we can train a premise selector (Balog et al., 2016) to predict library function usage given properties. Specifically, from the remaining properties, we

---

[7]Although we write $f$ as a function, that is, as returning an output, it is easy to handle procedures that do not return a value by defining $\tau_{out}$ to be a special void type.

compute estimated property signatures for each function in the training set, based on its input output examples. Then we use the property signature as the input to a feedforward network that predicts the number of times each library function appears in the program. In Section 4, we will give more details about the architecture of this premise selector, and evaluate it for synthesis. For now, we point out that this premise selector could itself be used to find useful properties, by examining which properties are most useful for the model's predictions.

### 3.3 WHY ARE PROPERTY SIGNATURES USEFUL?

Experiments in the next section will establish that property signatures let our baseline synthesizer emit programs it previously could not, but we think that they can have broader utility:

- They allow us to represent more types of functions. Property signatures can automatically deal with unbounded data types, recursive data types, and polymorphic functions.

- They reduce dependency on the distribution from which examples are drawn. If the user of a synthesizer gives example inputs distributed differently than the training data, the 'estimated' properties might not change much.[8]

- They can be used wherever we want to search for functions by semantics. Imagine a search engine where users give a specification, the system guesses a property signature, and this signature guess is used to find all the pre-computed functions with similar semantics.

- Synthesized programs can themselves become new properties. For example, once I learn a program for primality checking, I can use primality checking in my library of properties.

## 4 PROGRAM SYNTHESIS WITH PROPERTY SIGNATURES

We design an experiment to answer the following question: Can property signatures help us synthesize programs that we otherwise could not have synthesized? As we will show, the answer is yes!

### 4.1 EXPERIMENTAL SETUP

**How Does the Baseline Synthesizer Work?** Our baseline synthesizer is very similar to that in Feser et al. (2015) and works by filling in typed holes[9]. That is, we infer a program type $\tau_{in} \to \tau_{out}$ from the specification and the synthesizer starts with a empty 'hole' of type $\tau_{in} \to \tau_{out}$ and then fills it in all possible ways allowed by the type system. Many of these ways of filling-in will yield new holes, which can in turn be filled by the same technique. When a program has no holes, we check if it satisfies the spec. We order the programs to expand by their cost, where the cost is essentially a sum of the costs of the individual operations used in the program.

At the beginning of the procedure, the synthesizer is given a configuration, which is essentially a weighted set of *pool elements* that it is allowed to use to fill in the holes. A pool element is a rewrite rule that replaces a hole with a type-correct Searcho program, which may itself contain its own, new holes. In our synthesizer, there is one possible pool element for each of the 86 library functions in Searcho, which calls the library function, with correctly-typed holes for each of its arguments. The configuration will specify a small subset of these pool elements to use during search. It is through the configuration that we will use machine learning to inform the search procedure, as we describe later. See Appendix A.1 for further details on this baseline system.

**How is the Training Data Generated?** Our test corpus contains programs with 14 different types. For each of those 14 types, we randomly sample configurations and then randomly generate training programs for each configuration, pruning for observational equivalence. We generate up 10,000

---

[8]This argument does rely on properties being somehow simple. For instance, if the property does not compute whether a list contains the value 777, it cannot fail to generalize with respect to the presence or absence of 777. Since we search for properties in a shortest-first fashion, the properties we find should be biased toward simplicity, though certainly this hypothesis merits more experimental validation.

[9]In the synthesis literature, this approach of first discovering the high-level structure and then filling it in is sometimes called 'top-down' synthesis (Solar-Lezama, 2018). Top-down synthesis is to be contrasted with 'bottom-up' synthesis, in which low-level components are incrementally combined into larger programs.

semantically distinct programs for each type, though of course some function types admit less distinct programs than this ( e.g. Bool $\rightarrow$ Bool). We also generate and prune random properties as described in Section 3.2. See Listing 5 for examples of useful properties that were generated.

```
1  \:(List<Int>, List<Int>)->Bool (input, output) {
2    list_for_all_<Int> (input, \x {in_list_<Int> (x, output)})}
3  \:(List<Int>, List<Int>)->Bool (input, output) {
4    not_ (is_even_ (list_len_<Int> output))}
5  \:(List<Int>, List<Int>)->Bool (input, output) {
6    not_equal_<Int> ((ints_sum_ input), (ints_sum_ output))}
7  \:(List<Int>, List<Int>)->Bool (input, output) {
8    gt_ ((list_len_<Int> input), (list_len_<Int> output))}
```

Listing 5: 4 of the Properties with the highest discriminative power on functions of type List<Int> $\rightarrow$ List<Int>. The first checks if every element of the input list is in the output list. The second checks if the length of the output list is even. The third checks if sum of the input and the output list is the same, and the fourth checks if the input list is longer than the output list.

**How was the Test Set Constructed?**   We've constructed a test set of 185 human generated programs ranging in complexity from one single line to many nested function calls with recursion. Programs in the test set include computing the GCD of two integers, computing the $n$-th fibonacci number, computing the intersection of two sets, and computing the sum of all pairs in two lists. We ensure that none of the test functions appear in the training set. See the open source code for more details on this.

**What is the Architecture of the Model?**   As mentioned above, we train a neural network to predict the number of times each pool element will appear in the output. This neural network is fully connected, with learned embeddings for each of the values AllTrue, AllFalse and Mixed.

**How does the Model Output Inform the Search Procedure?**   Since we have a large number of pool elements (86), we can't run the synthesizer with all pool elements if we want to find programs of reasonable length. This is both because we will run out of memory and because it will take too long. Thus, we randomly sample configurations with less pool elements. We then send multiple such configurations to a distributed synthesis server that tries them in parallel.

When we use the model predictions, we sample pool elements in proportion to the model's predicted number of times that pool element appears. The baseline samples pool elements in proportion to their rate of appearance in the training set.

### 4.2 USING PROPERTY SIGNATURES LETS US SYNTHESIZE NEW FUNCTIONS

We ran 3 different runs of our distributed synthesizer for 100,000 seconds with and without the aid of property signatures. The baseline synthesizer solved 28 test programs on average. With property signatures, the synthesizer solved an average of 73 test programs. See Figure 1 for more discussion. Indeed, it can be seen from the figure that not only did the synthesizer solve many more test programs using property signatures, but it did so much faster, synthesizing over twice as many programs in one-tenth of the time as the baseline.

### 4.3 COMPARISON WITH DEEPCODER

We have conducted an experiment to compare premise selection using Property Signatures to the premise selection algorithm from (Balog et al., 2016). This required considerable modifications to the experimental procedure.

First, since the premise-selection part of DeepCoder can only handle Integers and lists of Integers, we restricted the types of our training **and** test functions. In particular, we read through (Balog et al., 2016) and found four function types in use:

```
1  f :: [Int] -> [Int]
2  g :: [Int] -> Int
3  h :: ([Int], [Int]) -> Int
4  k :: ([Int], Int) -> Int
```

Listing 6: The four function types used in DeepCoder.

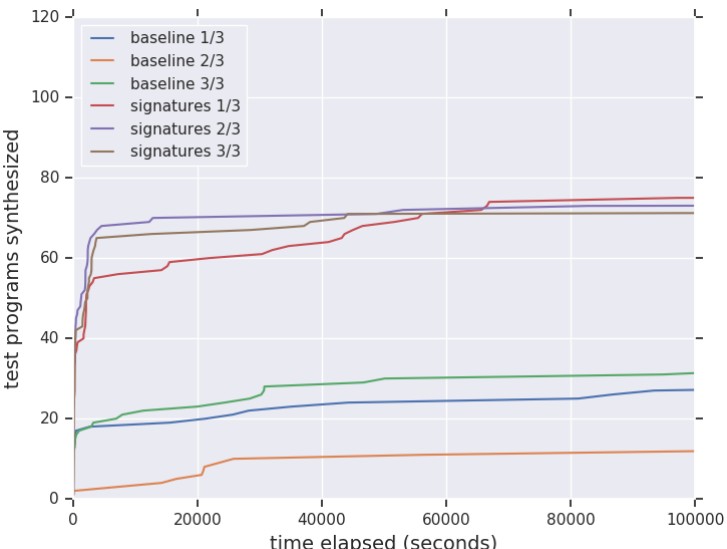

Figure 1: Comparison of synthesis with property signatures and without property signatures. The $x$-axis denotes time elapsed in seconds. Roughly speaking, we let the distributed synthesizer run for 1 day. The $y$-axis represenets the cumulative number of programs synthesized. On average, the baseline solved 28 of the test programs, while the baseline enhanced with property signatures solved 73 test programs (around 2.6 times as many programs). Both the baseline and the run with property signatures were run with three different random seeds. Altogether, this experiment provides strong evidence that property signatures can be useful.

The types of f and g in 6 are taken directly from (Balog et al., 2016). The types of h and k are inferred from examples given in the appendix of (Balog et al., 2016). Their DSL does not technically have tuples, but we have wrapped the inputs of their 'two-input-functions' in tuples for convience.

Second, since DeepCoder can only handle integers betwen $-255$ and $255$, we first re-generated all of our random inputs (used for 'hashing' of generated training data) to lie in that range. We then generated random training functions of the above four types. We then made a data set of training functions associated with 5 input-output pairs, throwing out pairs where any of the outputs were outside the aforementioned range, and throwing out functions where *all* outputs contained some number outside that range.

Third, of the examples in our test set with the right types, we modified their input output pairs in a similar way. We filtered out functions that could not be so modified. After doing so, we were left with a remaining test suite of 32 functions.

Finally, we trained a model to predict functions-to-use from learned embeddings of the input-output pairs, as in DeepCoder. We didn't see a description of how functions with multiple inputs had their inputs embedded, so we elected to separate them with a special character, distinct from the null characters that are used to pad lists.

Compared with the Property Signatures method, this technique results in far fewer synthesized test set programs. We did 3 random restarts for each of DeepCoder, Property Signatures, and the Random Baseline (recall that the random baseline itself is already a relatively sophisticated synthesis algorithm - it's just the configurations that are random). The 3 DeepCoder runs synthesized an average of 3.33 test programs, while the Property Signature runs (trained on the same modified training data and tested on the same modified test data) synthesized 16.33. The random baseline synthesized 3 programs on average.

A priori, this seems like a surprisingly large gap, but it actually fits with what we know from existing literature. Shin et al. (2018) observe something similar: which is that DeepCoder-esque techniques tend to generalize poorly to a a test set where the input-output pairs come from a different distribution than they do in training. This is the case in our experiment, and it will be the case in any realistic

setting, since the test set will be provided by users. Property Signatures are (according to our experiments) much less sensitive to such shift. This makes intuitive sense: whether an input list is half the length of an output list (for instance) is invariant to the particular distribution of members of the list.

Note that even if Property Signatures did not outperform DeepCoder on this subset of our test set, they would still constitute an improvement due to their allowing us to operate on arbitrary programs and inputs types.

## 5 PREDICTING PROPERTY SIGNATURES OF FUNCTION COMPOSITIONS

Most programs involve composing functions with other functions. Suppose that we are trying to solve a synthesis problem from a set of input/output examples, and during the search we create a partial program of the form $f(g(x))$ for some unknown $g$. Since we know $f$, we know its property signature. Since we have the program specification, we also have the estimated property signature for $f \circ g := f(g(x))$. If we could somehow guess the signature for $g$, we could look it up in a cache of previously computed functions keyed by signature. If we found a function matching the desired signature, we would be done. If no matching function exists in the cache, we could start a smaller search with only the signature of $g$ as the target, then use that result in our original search. We could attempt to encode the relationship between $f$ and $g$ into a set of formal constraints and pass that to a solver of some kind (De Moura & Bjørner, 2008), and while that is potentially an effective approach, it may be difficult to scale to a language like Searcho. Instead, we can simply train a machine learning model to predict the signature of $g$ from the signature of $f$ and the signature of $f \circ g$.

Here we present an experiment to establish a proof of concept of this idea. First, we generated a data set of 10,000 random functions taking lists of integers to lists of integers. Then we randomly chose 50,000 pairs of functions from this list, arbitrarily designating one as $f$ and one as $g$. We then computed the signatures of $f$, $g$ and $f \circ g$ for each pair, divided the data into a training set of 45,000 elements and a test set of 5,000 elements, and trained a small fully connected neural network to predict the signature of $g$ from the other two signatures.

On the test set, this model had 87.5% accuracy, which is substantially better than chance. We inspected the predictions made on the test set and found interesting examples like the one in Listing 7, where the model has learned to do something you might (cautiously) refer to as logical deduction on properties. This result is suggestive of the expressive power of property signatures. It also points toward exciting future directions for research into neurally guided program synthesis.

```
1 f: \:List<Int>->List<Int> inputs {
2   consume_ (inputs, (list_foldl_<Int, Int> (inputs, int_min, mod_)))}
3 g: \:List<Int>->List<Int> inputs {
4   list_map_<Int, Int> (inputs, neg_)}
5 prop: \:(List<Int>, List<Int>)->Bool (inputs, outputs) {
6   list_for_all_<Int> (outputs, \x {in_list_<Int> (x, inputs)})}
```

Listing 7: Example of successful prediction made by our composition predictor model. The property in question checks whether all the elements of the output list are members of the input list. For $f$, the value is AllTrue, and for $f \circ g$ the value is Mixed. The model doesn't know $g$ or its signature, but correctly predicts that the value of this property for $g$ must be Mixed.

## 6 RELATED WORK

There is substantial prior work on program synthesis in general. We can hardly do it justice here, but see some of Gottschlich et al. (2018); Solar-Lezama (2018); Gulwani et al. (2017); Allamanis et al. (2018) for more detailed surveys.

**Property Based Testing:** Function properties are similar to the properties from Property Based Testing, a software testing methodology popularized by the QuickCheck library (Claessen & Hughes, 2011) that has now spread to many contexts (Gallant, 2018; Holser, 2018; Hypothesis, 2018; Luu, 2015; Elhage, 2017; MacIver, 2017). Quickcheck properties are human-specified and operate on functions, while our properties operate on input/output pairs.

**Automated Theorem Proving:** Synthesizing programs using machine learning is related to the idea of proving theorems using machine learning (Irving et al., 2016). Synthesis and theorem proving are formally related as well (Howard, 1980).

**Program Synthesis from a Programming Languages Perspective:** Most existing work on synthesis approaches is from the perspective of programming language design. Our baseline synthesizer borrows many ideas from Feser et al. (2015). Polikarpova et al. (2016) use refinement types (Freeman, 1994) (roughly, a decidable version of dependent types - see Pierce & Benjamin (2002)) to give program specifications, allowing the type-checker to discard many candidate programs. Property signatures can be thought of as a compromise between refinement types and dependent types: we can write down specifications with them that would be impossible to express in refinement types, but we can only check those specifications empirically.

**ML-Guided Program Synthesis:** More recently, researchers have used machine learning to synthesize and understand programs. We have mentioned Balog et al. (2016), but see all of: Nye et al. (2019); Ellis et al. (2018a); Zohar & Wolf (2018); Kalyan et al. (2018); Ellis et al. (2019a); Liang et al. (2010); Alon et al. (2019) as well. Menon et al. (2013) introduces the idea of features: a predecessor to the idea of properties. Features differ from properties in that they are hand-crafted rather than learned, and that they were applied only on a limited string processing domain.

**Deepcoder:** The relationship between this work and Balog et al. (2016) merits special discussion. Aside from the inclusion of property signatures, they differ in the following ways:

- We use a more expressive DSL. Their DSL only allows linear control flow with a small set of functions, whereas our language is Turing complete (it has looping, recursion, etc). We also have a larger set of allowed component functions: 86 vs. 34.

- Their machine learning method does not work straightforwardly for arbitrary programs. Their training and test programs only deal with integers and lists of integers, while we have 14 different function types. It would thus not be feasible to compare the techniques on anything but a tiny subset of our existing test set.

- The test cases in Balog et al. (2016) are generated from their enumerative synthesizer. It is therefore guaranteed that the synthesizer will be able to emit them in a reasonable amount of time during testing, so their demonstrated improvements are 'merely' speed-ups. Our test cases are human generated, and over half of the programs synthesized using property signatures were not synthesized at all[10] given over a day of time.

## 7 CONCLUSION AND FUTURE WORK

In this work, we have introduced the idea of properties and property signatures. We have shown that property signatures allow us to synthesize programs that a baseline otherwise was not able to synthesize, and have sketched out other potential applications as well. Finally, we have open sourced all of our code, which we hope will accelerate future research into ML-guided program synthesis.

ACKNOWLEDGMENTS

We would like to thank Kensen Shi, David Bieber, and the rest of the Program Synthesis Team for helpful discussions. We would like to thank Colin Raffel for reading a draft of the paper. Most of all, we owe a substantial debt to Niklas Een, on whose Evo programming language (`https://github.com/tensorflow/deepmath/tree/master/deepmath/zz/CodeBreeder`) the Searcho language is heavily based.

---

[10] Of course, barring bugs in the synthesizer, they would be synthesized *eventually*.

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

**Data:** A PBE spec and a synthesizer configuration
**Result:** A program satisfying the specification (hopefully!)
Queue.push(hole :: $\tau_{in} \to \tau_{out}$);
**while** *Queue is not empty* **do**

> partial_program ← GetLowestCostPartial(*Queue*);
> **if** HasHoles(*partial_program*) **then**
> > ExpandOneHole(*partial_program*);
>
> **end**
> **else**
> > TestAgainstSpec(*partial_program*);
>
> **end**

**end**

Figure 2: The top-down synthesizer that we use as a baseline in this work. In a loop until a satisfying program is found or we run out of time, we pop the lowest-cost partial program from the queue of all partial programs, then we fill in the holes in all ways allowed by the type system, pushing each new partial program back onto the queue. If there are no holes to fill, the program is complete, and we check it against the spec. The cost of a partial program is the sum of the costs of its pool elements, plus a lower bound on the cost of filling each of its typed holes, plus the sum of the costs of a few special operations such as tuple construction and lambda abstraction.

## A APPENDIX

### A.1 FURTHER DETAILS ON THE BASELINE SYNTHESIZER

This section contains details on the baseline synthesizer that did not fit into the main text. Figure 2 gives a more formal description of the basic synthesis algorithm. Listing 8 shows an example trajectory of partial program expansions.

```
1 $1 \:(Int, Int)->(Int, Int) (a2, a3) {?}
2 $2 \:(Int, Int)->(Int, Int) (a2, a3) {(?, ?)}
3 $2 \:(Int, Int)->(Int, Int) (a2, a3) {(a3, ?)}
4 $2 \:(Int, Int)->(Int, Int) (a2, a3) {(a3, a2)}
```

Listing 8: The trajectory the synthesizer took to generate the swap function, which just swaps the two elements of a tuple. Since it knows it needs to take a tuple of ints as an argument and return a tuple of ints, it starts with a hole of type (Int, Int) in line 1. It then converts that hole into a tuple of holes, both of type Int in line 2, fills one of the holes with a reference to one of the arguments in line 3, and fills in the final hole with a reference to the other argument in line 4. Note that this listing doesn't show all programs attempted, it just shows the sequence of partial programs that led to the final solution.

