# OpenReview forum: "Learning to Represent Programs with Property Signatures"
_ICLR.cc/2020/Conference — Accept (Poster)_

### Official Review · AnonReviewer3 · 2019-10-25
**Official Blind Review #3**

**Rating:** 6

**Review:**

I would like to be able to recommend accepting this paper, but I can't.  It describes two contributions to the community that could be valuable:

1: searcho, a programming language designed for studying program synthesis via search over programs
2: A method of automatically choosing interesting features for guided program search.

The paper does not give enough evidence to ascertain the value of either contribution.  There are now a large number of program synthesis works using ML, and a huge literature on program synthesis without ML.   While many of ML works use a DSL as the testbed, surely the authors' feature selection method can be applied in some of these DSLs, allowing comparisons with current art?   On the other hand, many of the ML methods don't require the DSL used in the works that introduce them.  Can searcho + your set of training and test programs distinguish between these methods, and establish a benchmark?  Are there some more realistic tasks/applications that searcho makes reasonable, not readily approached with previous DSLs used for testing these algorithms?   I think this paper could be valuable if it could demonstrate either in what ways the new feature selection algorithm is  an improvement over prior ML methods, or that searcho is valuable as a benchmark or for approaching interesting applications.


######################################################################################
edit after author response:  raising to 6, I think the deepcoder experiments are useful in contextualizing the contribution of the authors' feature selection method.



**Experience Assessment:**

I have read many papers in this area.

**Review Assessment: Checking Correctness Of Derivations And Theory:**

I assessed the sensibility of the derivations and theory.

**Review Assessment: Checking Correctness Of Experiments:**

I assessed the sensibility of the experiments.

**Review Assessment: Thoroughness In Paper Reading:**

I read the paper at least twice and used my best judgement in assessing the paper.

---

> ### Author Response · Authors · 2019-11-06
> **Thanks for the review!**
>
> Thanks very much for the review.
> Please see our comment titled OVERALL RESPONSE regarding comparisons to prior work.
> Through a combination of arguments made here and the additional experiments we are running, we hope to convince you that a slightly higher score is merited.
> In this comment, we will use > for quotes and respond point-by-point.
>
> > huge literature on program synthesis without ML.
> Agreed, but since our baseline is essentially a (faster and more general) implementation of [1],
> we would claim that we have made a serious effort to compare to work from this literature.
> Do you find this claim unsatisfying?
>
> > On the other hand, many of the ML methods don't require the DSL used in the works that introduce them.
> Point taken, and as mentioned in our other comment, we are setting up an experiment right now that will
> compare premise-selection with property signatures to premise-selection using the algorithm from [2].
> Please let us know if there is some other comparison you'd like to see and we will do our best.
>
> *However*, we would argue that there are several important ways in which this work adds to current art
> that can be verified without actually running experiments in the way that is common in the ML literature.
> That is, there are things that our system can do 'by construction' that previous ML-based methods can't do.
> For example, DEEPCODER (the most relevant prior art, probably?)
>
> a) Has no way to deal with unbounded data types,
> b) Requires hard-coding a new neural network architecture for all compound data-types,
> c) Has no way to deal with polymorphic functions
>
> and Property Signatures deal with all of these things 'for free'.
> To be (just a little bit) glib, there's a sense in which we have run a comparison
> with DeepCoder already (on almost all of the 13 types we used), and it just 'returned NaN' when
> there was no way for it to handle the type under consideration.
> Again, we are actually going to run the experiments for the few types where it makes sense and report back,
> but we would claim that your review as it currently stands does not give us enough credit for points like
> a), b) and c) above.
>
> > Are there some more realistic tasks/applications that searcho makes reasonable, not readily approached with previous DSLs used for testing these algorithms?
> We would argue that the answer is yes, because Searcho is Turing complete and comes w/ a distributed search implementation, but we might
> be misunderstanding you on this point?
>
> Finally, we are pretty optimistic about the value of the ideas in Section 5, but it doesn't seem like any of the reviewers found that section interesting.
> We would be especially grateful to have more specific feedback on that section.
>
> [1] Synthesizing data structure transformations from input-output examples
>   ( https://www.cs.rice.edu/~sc40/pubs/pldi15.pdf )
> [2] DEEPCODER: LEARNING TO WRITE PROGRAMS
>   ( https://openreview.net/pdf?id=ByldLrqlx )

---

> ### Author Response · Authors · 2019-11-13
> **New Experiment**
>
> Hi and thanks again for the review!
>
> We believe we've addressed (or at least made substantial progress toward addressing) your main criticism:
> Please see our comment titled: 'New Experiment: Comparison with DeepCoder' and let us know if you have
> any questions about that comment or about our original response.
> We would be happy to expand on them or run additional comparisons as time permits.

---

### Official Review · AnonReviewer2 · 2019-10-25
**Official Blind Review #2**

**Rating:** 6

**Review:**

** Summary
The paper studies the problem of program synthesis from examples and it proposes the notion of property signature as a set of "features" describing the program that can be used to simplify the synthesis by combining together sub-programs with similar features.

** Evaluation
The paper is outside my scope of research, so my review is an educated guess.

The use of properties to summarize some features about the program and the possibility to evaluate them directly from samples seems a very interesting idea. As the authors mentioned, this may help in creating useful features and useful architectures to simplify the learning task. The concept of property signatures is very well introduced and explained. The authors also provide an extensive comparison to related work. The empirical results seem promising in showing how property signatures make the synthesis much faster and better.

The downsides of the paper are:
- While it is clear how to build property signatures, it is quite unclear to me how they simplify the generation of programs that combine smaller/simpler programs.
- Sect 3.2 on how to learn useful properties is rather vague and it would need a much more detail explanation.
- Although the authors release code for the paper, the description of the experiments seem rather shallow and it requires a much higher level of detail on how the learning process is set up and executed.


**Experience Assessment:**

I do not know much about this area.

**Review Assessment: Checking Correctness Of Derivations And Theory:**

N/A

**Review Assessment: Checking Correctness Of Experiments:**

I assessed the sensibility of the experiments.

**Review Assessment: Thoroughness In Paper Reading:**

I read the paper at least twice and used my best judgement in assessing the paper.

---

> ### Author Response · Authors · 2019-11-06
> **Thank you for the review!**
>
> Thanks very much for the review!
>
> We will use > for quotes and respond point-by-point below:
>
> > it is quite unclear to me how they simplify the generation of programs that combine smaller/simpler programs
> We will expand the explanation of this part in the text.
>
> > Sect 3.2 on how to learn useful properties is rather vague and it would need a much more detail explanation.
> You're right - we will fix this.
>
>
> > the description of the experiments seem rather shallow
> Yeah, we were really fighting with the space restrictions here.
> We will expand the description substantially and move things to an appendix as necessary.

---

### Official Review · AnonReviewer1 · 2019-11-02
**Official Blind Review #1**

**Rating:** 1

**Review:**

This paper proposed the concept of "property signatures" , which are learned to represent programs. The property signatures are essentially some key attributes that one may summarize from a given set of input-output pairs, which the target function has. Then a program can be generated by evaluating these property signatures vectors (which is simply a bag-of-word representation with only 0 or 1 as each element). Much discussions have been given to discuss why and how these properties may be useful and very little real experiments are conducted quantitatively compared with existing works. Although this paper is quite interesting, I think this paper is in its very early stage and there are a lot of serious concerns I have for using this approach to synthesize the real complex programs.

1) First of all, the notion of property signatures are easy to understand and is very natural. Just like human beings, when we write a program, we first think about the possible attributes of this program may have given a set of input-output pairs for both correctness and completeness. However, this is also the hard part of this idea. Potentially it could have an exponential number of possible properties as the program goes more complex and complex. It will quickly become computationally intractable problem.

2) When I read the middle of paper, I would eager to know how authors can effectively find a good set of properties of a target program from a given input-output pairs. However, when I eventually reached the Section 4, I was kindly disappointed since I did not see any effective and principle way to get them. All I saw are "randomly sample and generate". This may be Ok for a very simple program given a set of simple input-output pairs. But it is definitely not feasible for any complex function, not to mention project. I think this is the key for the proposed idea since how to construct a good set of property signatures is crucial to treat them as the inputs for any program synthesis task later.

3) There are very little baselines to compare against even though authors listed "substantial prior work on program synthesis". I understand the existing works may have their limitation in both what they can do and how well they can do. But it is still important to compare with directly on the same set of benchmarks. Otherwise, it is hard to be convincing that this approach is indeed superior compared to existing ones.

**Experience Assessment:**

I have read many papers in this area.

**Review Assessment: Checking Correctness Of Derivations And Theory:**

I carefully checked the derivations and theory.

**Review Assessment: Checking Correctness Of Experiments:**

I carefully checked the experiments.

**Review Assessment: Thoroughness In Paper Reading:**

I read the paper thoroughly.

---

> ### Author Response · Authors · 2019-11-06
> **Thanks for the review!**
>
> Thanks very much for the review!
> Please see our comment titled OVERALL RESPONSE, regarding your point 3.
> Overall we believe that there are several misunderstandings (our fault for not being more clear).
> We hope that you will consider raising your score slightly in response to our clarifications.
>
> We will quote using > and respond point by point:
>
> ====================    Intro   ==============================
> > which is simply a bag-of-word representation with only 0 or 1 as each element
> This is not quite true: there are 3 possible elements: ALL_TRUE, ALL_FALSE, or MIXED.
> You could think of them as -1, 0 and 1.
> We don't mean to be pedantic, but we think this is an important point and possibly points to a larger misunderstanding:
> The value for a given property applies across all possible input/output pairs of the relevant type.
> So if you have the property 'is the output list twice the length of the input list', it will be MIXED for the function
> that concatenates a list to itself, since that function takes the empty list to the empty list, but it takes [1,2,3] -> [1,2,3,1,2,3].
> Apologies if you already understood this.
>
> ====================    Point 1) ==============================
> > Potentially it could have an exponential number of possible properties
> We think there is a misunderstanding here.
> For most type signatures, there are in fact infinitely many possible properties, since properties are just programs.
> But there's no need to actually use them all: think of property signatures as analogous to random projections.
> We get a decent representation of the semantics of a program from a small-ish number of property signatures.
>
> ====================    Point 2) ==============================
> > All I saw are "randomly sample and generate".
> This misses a few really important parts.
> First, we aren't just randomly sampling from all possible properties, we're sampling
> lowest-cost-first, which means that we will get simple properties before more complicated properties.
> Second, there's a filtering stage that you may have missed?
> Given a test set of program specifications, you can just only keep properties
> that help you to distinguish between multiple test specifications.
>
> > But it is definitely not feasible for any complex function, not to mention project.
> This is a very strong assertion.
> It depends what you mean by complex, but we would argue that the following function
> is pretty complex (from Listing 1 in the paper):
>
> fun unique_justseen(xs :List<Int>) -> List<Int> {
>   let triple = list_foldl_<Int, (List<Int>, Int, Bool)>(
>     xs,
>     (nil<Int>, 0, _1),
>     \(list_elt, (acc, last_elt, first)){
>       cond_(or_(first, not_equal_(list_elt, last_elt)),
>         \{(cons_(list_elt, acc), list_elt, _0)},
>         \{(acc , list_elt, _0)})
>     });
>   list_reverse_(#0(triple))
> };
>
> And we were able to learn to synthesize this function.
>
> More generally, it's probably wrong to envision program synthesis tools
> that synthesize whole projects from scratch: it seems much more likely that
> users of future tools will synthesize projects function-by-function,
> in which case your concern would not apply.
>
> ====================    Point 3) ==============================
> > There are very little baselines to compare against
> Please see our comment titled OVERALL RESPONSE for this.
> We should have been more clear, but our main baseline is in fact an implementation of prior work.

---

> ### Author Response · Authors · 2019-11-13
> **New Experiment**
>
> Hi and thanks again,
>
> We've run a new experiment that we believe largely addresses your 3rd point.
> We wonder if, in light of this new experiment and our previous response (which addresses your 1st and 2nd points),
> you might consider increasing your score slightly?
> In light of the other two reviews, a score of 1 seems perhaps a bit harsher than is warranted?
>
> Please let us know if there are any other questions we can answer.

---

### Author Response · Authors · 2019-11-06
**OVERALL RESPONSE: We *do* compare to prior work (but we will add even more such comparisons)**

Thanks for the reviews!

We'll respond to each reviewer in detail, but here we address one main point
that was mentioned by Reviewers 1 and 3:

Reviewers 1 and 3 were concerned about a lack of comparison to prior work:
We should have made this more clear, but our main experiment *does* already have a comparison to prior work,
since the baseline synthesizer is essentially the algorithm from [1].
Whether this represents the 'state-of-the-art' is up for debate, since synthesis papers tend to all use
very different benchmarks, but we don't know of anything that has obviously superseded it?

Regardless, we are happy to add more such comparisons, and are currently working on a comparison
with the 'DeepCoder' algorithm from [2].
We will add the results here before the end of the reviewer response period.
Please let us know if there are other comparisons you would like to see.

[1] Synthesizing data structure transformations from input-output examples
  ( https://www.cs.rice.edu/~sc40/pubs/pldi15.pdf )
[2] DEEPCODER: LEARNING TO WRITE PROGRAMS
  ( https://openreview.net/pdf?id=ByldLrqlx )

---

### Author Response · Authors · 2019-11-13
**New Experiment: Comparison with DeepCoder**

We have conducted an experiment to compare premise selection using Property
Signatures to the premise selection algorithm from [1].
This required considerable modifications to the experimental procedure:

First, since the premise-selection part of DeepCoder can only handle Integers
and lists of Integers, we restricted the types of our training *and* test
functions. In particular, we read through [1] and found four
function types in use:

f :: [Int] -> [Int]
g :: [Int] -> Int
h :: ([Int], [Int]) -> Int
k :: ([Int], Int) -> Int

The types of f and g  are taken directly from
[1]. The types of h and k are inferred from examples given in the
appendix of [1]. Their DSL does not technically have tuples,
but we have wrapped the inputs of their 'two input functions' in tuples for
convenience.

Second, since DeepCoder can only handle integers between -255 and 255, we
first re-generated all of our random inputs (use for 'hashing' of generated
training functions) to lie in that range. We then generated random training functions
of the above four types. We then made a data set of training functions
associated with 5 input-output pairs,
throwing out pairs where any of the outputs
were outside the aforementioned range, and throwing out functions where
*all* outputs contained some number outside that range.

Third, of the examples in our test set with the right types, we modified their
input output pairs in a similar way.
We filtered out functions that could not be so modified.
After doing so, we were left with a remaining test suite of 32 functions.

Finally, we trained a model to predict functions-to-use from learned embeddings
of the input-output pairs, using the architecture described in [1].
We didn't see a description of how functions with multiple inputs had their
inputs represented, so we elected to separate them with a special character,
distinct from the null characters that are used to pad lists.

Compared with the Property Signatures method, this technique results in far
fewer synthesized test set programs.
We did 3 random restarts for each of DeepCoder, Property Signatures, and the
Random Baseline (recall that the random baseline itself is already a relatively
sophisticated synthesis algorithm - it's just the configurations that are
random).
The DeepCoder runs synthesized an average
of 3.33 test programs, while the Property Signature runs (trained on the same
modified training data and tested on the same modified test data) synthesized
16.33. The random baseline synthesized 3 programs on average.

A priori, this might seem like a surprisingly large gap, but it actually fits with
what we know from existing literature.
It's well known that deep learning techniques perform poorly under distribution shift in general,
and [2] observe something similar for program synthesis:
which is that DeepCoder-esque techniques tend to generalize poorly to a
a test set where the input-output pairs come from a different distribution
than they do in training.
This is the case in our experiment, and it will be the case in any realistic
setting, since the test set will be provided by users of the synthesis tool.
Property Signatures are (according to our experiments) much less sensitive to such shift.
This makes intuitive sense: whether an input list is half the length of an output list (for instance) is
invariant to the particular distribution of members of the list.

Note that even if Property Signatures did not outperform DeepCoder on this
subset of our test set, they would still constitute an improvement due to their
allowing us to operate on arbitrary programs and input types.


[1] DEEPCODER: LEARNING TO WRITE PROGRAMS
  ( https://openreview.net/pdf?id=ByldLrqlx )

[2] Synthetic Datasets for Neural Program Synthesis
  ( https://openreview.net/forum?id=ryeOSnAqYm )

PS:
We are updating the draft to include these results, but we thought it would be easier
to follow a comment here than to try and see the difference between 2 PDFs.

---

### Author Response · Authors · 2019-11-15
**Revision**

We've added a description of the DeepCoder comparison experiment to the appendix.

We plan to move it into the main text after some wrangling.

---

### Decision · Program_Chairs · 2019-12-19

**Decision:**

Accept (Poster)

**Comment:**

The authors propose improved techniques for program synthesis by introducing the idea of property signatures. Property signatures help capture the specifications of the program and the authors show that using such property signatures they can synthesise programs more efficiently.

I think it is an interesting work. Unfortunately, one of the reviewers has strong reservations about the work. However, after reading the reviewer's comments and the author's rebuttal to these comments I am convinced that the initial reservations of R1 have been adequately addressed. Similarly, the authors have done a great job of addressing the concerns of the other reviewers and have significantly updated their paper (including more experiments to address some of the concerns). Unfortunately R1 did not participate in subsequent discussions and it is not clear whether he/she read the rebuttal. Given the efforts put in by the authors to address different concerns of all the reviewers and considering the positive ratings given by the other two reviewers I recommend that this paper be accepted.

Authors,
Please include all the modifications done during the rebuttal period in your final version. Also move the comparison with DeepCoder to the main body of the paper.